# Chiral evasion and stereospecific antifolate resistance in *Staphylococcus aureus*

**Siyu Wang**[1,2], **Stephanie M. Reeve**[3], **Graham T. Holt**[1,2], **Adegoke A. Ojewole**[1,2,¤], **Marcel S. Frenkel**[4], **Pablo Gainza**[1], **Santosh Keshipeddy**[3], **Vance G. Fowler**[5], **Dennis L. Wright**[3,6], **Bruce R. Donald**[1,4,7,8]*

**1** Department of Computer Science, Duke University, Durham, North Carolina, United States of America, **2** Program in Computational Biology and Bioinformatics, Duke University, Durham, North Carolina, United States of America, **3** Department of Pharmaceutical Sciences, University of Connecticut, Storrs, Connecticut, United States of America, **4** Department of Biochemistry, Duke University Medical Center, Durham, North Carolina, United States of America, **5** Division of Infections Diseases, Department of Medicine, Duke University Medical Center, Durham, North Carolina, United States of America, **6** Department of Chemistry, University of Connecticut, Storrs, Connecticut, United States of America, **7** Department of Mathematics, Duke University, Durham, North Carolina, United States of America, **8** Department of Chemistry, Duke University, Durham, North Carolina, United States of America

élions These authors contributed equally to this work.
¤ Current address: Genentech, Inc., San Francisco, California, United States of America
* brd+pcb21@cs.duke.edu

**Data Availability Statement:** The crystallography, atomic coordinates, and structure factors have been deposited in the Protein Data Bank, https://www.rcsb.org/ (PDB ID codes 7T7Q and 7T7S). All

## Abstract

Antimicrobial resistance presents a significant health care crisis. The mutation F98Y in *Staphylococcus aureus* dihydrofolate reductase (SaDHFR) confers resistance to the clinically important antifolate trimethoprim (TMP). Propargyl-linked antifolates (PLAs), next generation DHFR inhibitors, are much more resilient than TMP against this F98Y variant, yet this F98Y substitution still reduces efficacy of these agents. Surprisingly, differences in the enantiomeric configuration at the stereogenic center of PLAs influence the isomeric state of the NADPH cofactor. To understand the molecular basis of F98Y-mediated resistance and how PLAs' inhibition drives NADPH isomeric states, we used protein design algorithms in the OSPREY protein design software suite to analyze a comprehensive suite of structural, biophysical, biochemical, and computational data. Here, we present a model showing how F98Y SaDHFR exploits a different anomeric configuration of NADPH to evade certain PLAs' inhibition, while other PLAs remain unaffected by this resistance mechanism.

## Author summary

Antimicrobial resistance is a major healthcare crisis. While we were developing novel enzyme inhibitors to combat methicillin-resistant *Staphylococcus aureus* (MRSA), we found that the chirality of both inhibitor and cofactor can have a large influence on inhibitor potency. Our detailed study of enantiomeric propargyl-linked antifolates (PLAs) shows that the chiral state of inhibitors can affect the chiral state of the cofactor. Moreover, the bacterial enzyme target can exploit cooperative chirality to evade inhibitor binding. We call this phenomenon chiral evasion. Using crystal structures, biochemical assays,

of the computational results and code used and discussed in this manuscript are available from the Harvard Dataverse repository (https://dataverse.harvard.edu/dataset.xhtml?persistentId=doi:10.7910/DVN/5AQP7Z). For new empirical designs we recommend using the latest version of OSPREY available for free at http://www.cs.duke.edu/donaldlab/osprey.php. All computer code for the OSPREY system is also available on GitHub at https://github.com/donaldlab/OSPREY3, and is open-source and free.

**Funding:** We gratefully acknowledge grant support from the NIH (R01 GM078031, R01 GM118543, and R35 GM144042 to B.R.D., and R01 AI111957 to D.L.W.). The funders had no role in study design, data collection and analysis, decision to publish, or preparation of the manuscript.

computational protein design algorithms, and statistical mechanics, a detailed mechanism for chiral evasion is proposed. While the concept that different enantiomers have different biology is well known, MRSA is unique: we do not know of any other cases where a single mutation (F98Y) flips the chirality preference for cofactor binding and induces stereospecificity for drug binding. Thus, we illuminate the effect of this clinically relevant resistance mutation on the obligate cofactor binding site. These new insights will be useful to develop more durable antibiotics that are resilient to resistance.

## Introduction

Methicillin-resistant *Staphylococcus aureus* (MRSA) is one of leading causes of healthcare associated infections, pneumonia and skin and soft tissue infections (SSTIs). Antifolates such as trimethoprim (TMP) exhibit potent activity as antibacterial agents against many MRSA clinical isolates [1, 2]. Due to structural and sequence differences between human and prokaryotic dihydrofolate reductase (DHFR), antifolates can be developed as selective and safe antimicrobial agents [3]. Currently a fixed-dose combination of TMP and sulfamethoxazole, which targets dihydropteroate synthase, is one of the top ten oral antibiotics prescribed [4].

DHFR is an enzyme vital for cellular replication, mainly due to its important role in the one-carbon metabolism pathway [5]. DHFR reduces dihydrofolate (DHF) to tetrahydrofolate (THF) [6]. Nicotinamide adenine dinucleotide phosphate (NADPH) is required for this catalytic reaction. NADPH participates as a cofactor and donates a hydride to reduce C4 on the dihydropterin ring of DHF. Over the years, various models for different mechanisms of catalysis have been proposed [7–9]. During catalysis, NADPH adds a hydride to C6 of the dihydropterin ring and then N5 is protonated [6, 9]. In addition, in *Escherichia coli* DHFR, Tyr98 stabilizes the positive charge in the nicotinamide ring during the hydride transfer between C4 of the nicotinamide ring and C6 of the dihydropterin ring [10]. In wild type SaDHFR, position 98 is a phenylalanine. Its mutation to tyrosine confers resistance to antifolates such as TMP and some other novel antifolates according to experimental and clinical observations [11, 12].

New resistance mechanisms continue to emerge and challenge the efficiency of existing drugs [13]. Resistance to antifolates, including TMP, the only approved DHFR inhibitor for antibiotic use, has emerged in many cases [11, 14]. Therefore, second generation antifolates that can overcome resistance must be developed. This need motivates our studies to develop and apply rapid and effective computational methods to predict and understand the mechanisms of resistance.

Recently we have developed a series of propargyl-linked antifolates (PLA) that are potent SaDHFR inhibitors [12, 15]. As with TMP, PLAs are competitive inhibitors that compete with DHF to bind in the folate binding site of DHFR. The linear triple-bond linker makes it possible for PLAs to fit into the narrow binding pocket with limited steric clashes (see Figs 1 and 2a). PLAs also have a biaryl ring system that mimics the para-aminobenzoic acid (PABA) and glutamate moieties of folate. In contrast to traditional antifolates such as TMP, PLAs showed high potency against both wild type and TMP-resistant DHFR according to $IC_{50}$ measurement [12, 15].

In addition to enzymatic analysis, structure determination plays an important role in the study of resistance and development of PLAs. Crystal structures of various PLAs bound to SaDHFR (including both WT and F98Y variants) in complex with NADPH were determined to high resolution [12]. Interestingly, among all of crystal structures we determined in Ref. 12, four of them (PDB ID: 3FQF, 3FQO, 3FQV and 3FQZ) show two distinct bound forms of the

**Fig 1. Chemical structures of R-27, S-27, and trimethoprim (TMP).** The linear triple-bond linker of both R-27 and S-27 limits steric clashes within the binding pocket, and the biaryl ring system mimics the para-aminobenzoic acid (PABA) and glutamate moieties of folate [12, 15]. The chiral center of R-27 and S-27 is labeled with *(R)* and *(S)*, respectively.

NADPH cofactor. In addition to the common $\beta$-anomer of NADPH, these structures also contained what appeared to be a second alternative conformation of NADPH. However, subsequently in one of our drug resistance prediction studies for SaDHFR [16] using our open-source computational protein design (CPD) software OSPREY [17–19], we found that the alternative NADPH model was incorrectly assigned. We thus carefully refitted the NADPH based on the density maps of 3FQF, 3FQO, 3FQV, 3FQZ using COOT [20] and PHENIX [21] and performed extensive studies on analyzing the geometric features of the alternative NADPH analog. Over 4000 NADPH conformations binding to different proteins from various species among 1700 deposited PDB files were compared, and we determined that the alternative NADPH 'conformation' was in fact a diastereomer with a different anomeric configuration from the standard $\beta$ anomer NADPH [18, 22–26]. NADPH molecules with different anomeric configurations can naturally interconvert *in vitro* and both forms can be found *in vivo* [24].

The stereochemical configuration of molecules is an important factor to be considered in the context of biological processes, particularly in enzyme-catalyzing reactions since enzyme-substrate systems usually require strict chiral matching [27]. In the SaDHFR system, not only the NADPH cofactor but also the PLAs can bind in different configurations. In order to study how the chirality of the PLA influences inhibitor selectivity and DHFR stereospecificity, a series of PLA enantiomers (determined by a single chiral center on the propargyl linker) were synthesized and analyzed [15]. In general, most PLA enantiomers pairs can inhibit both WT and F98Y SaDHFR activity at low nanomolar concentration. Interestingly, some PLA enantiomer pairs showed substantially different potencies against SaDHFR. For example, the most potent PLA enantiomer pair R-27 and S-27, have nearly identical $IC_{50}$ values for WT SaDHFR, at 15 and 18 nM respectively. However against the F98Y mutation the R enantiomer suffers a 34-fold loss of potency ($IC_{50}$ 510 nM), whereas the S enantiomer suffers only a 6-fold loss of potency ($IC_{50}$ 111 nM) (Table 1). In other words, S-27 was found to be substantially more potent towards the F98Y substitution than R-27, despite having almost identical activity towards WT.

To analyze this phenomenon from a structural perspective and to study cofactor influence, two crystal structures of SaDHFR (WT) complexed with R-27/S-27 and NADPH (PDB ID: 4XEC and 4TU5) were solved [15]. According to these crystal structures, we observed that this pair of PLA enantiomers can selectively recruit different configurations of NADPH, which supported our analysis in [18]. S-27 binds together with $\beta$ form NADPH, while R-27 binds together with another type of NADPH isomer, with 100% occupancy in both cases (Fig 2a). The alternative NADPH isomer was initially assigned as $\alpha$-NADPH [15], but herein we identify it as a tricyclic NAPDH isomer (t-NADPH). NADPH is composed of ribosylnicotinamide 5′-phosphate coupled by a pyrophosphate linkage to the 5′-phosphate adenosine 2′,5′-diphosphate (PADP). The asymmetric center of NADPH discussed here is the 1′ anomeric carbon on

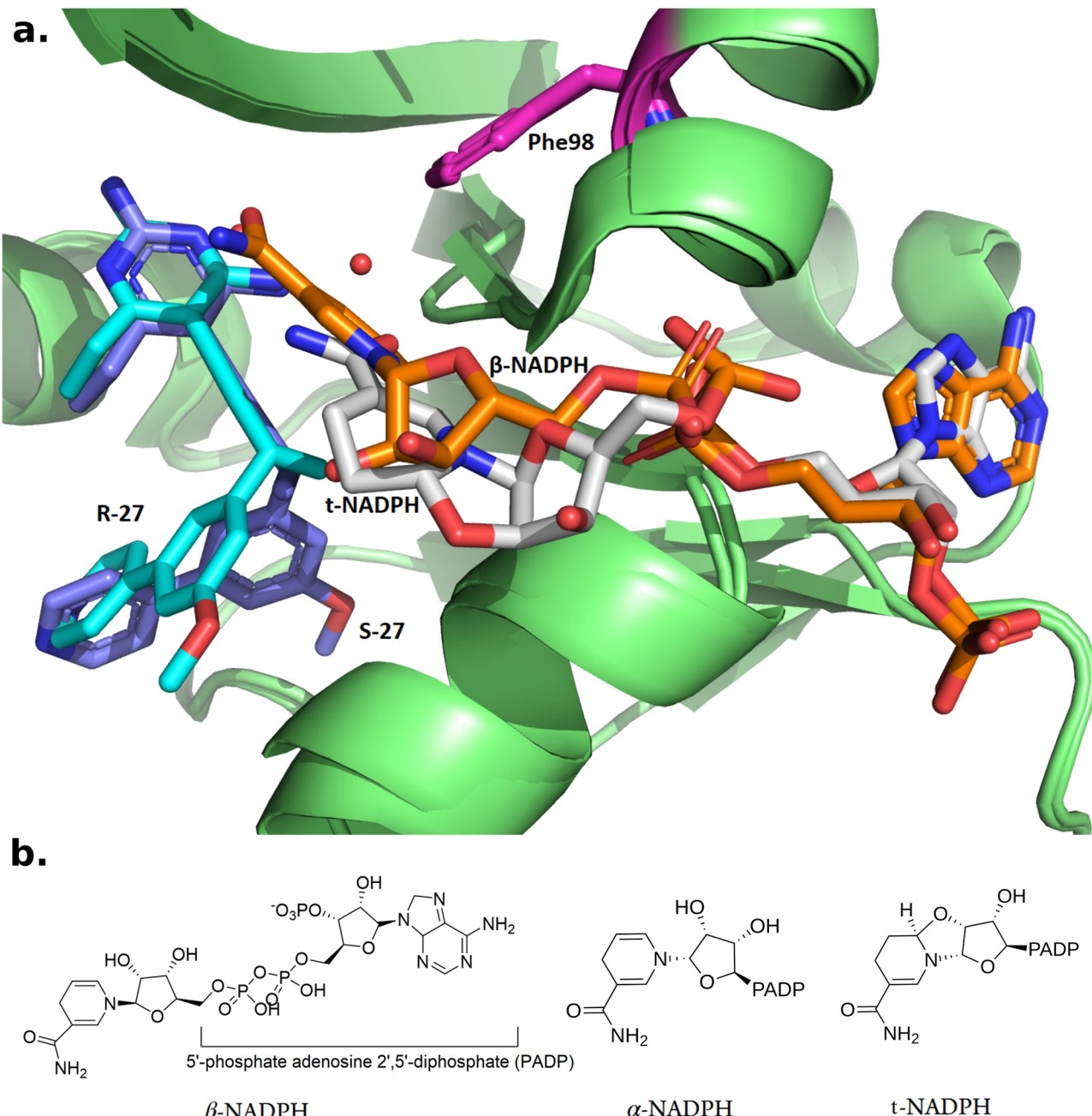

**Fig 2. Structure of different configurations of NADPH. a.** Crystal structures of R-27 (colored in cyan) bound to DHFR and t-NADPH (colored in light grey), and S-27 (purple) bound to DHFR and $\beta$-NADPH (orange). PDB entries are 7T7S and 4TU5. Phe98 is highlighted in magenta. Starting from the propargylic stereocenter, the biaryl moieties on R-27 and S-27 go to totally different directions and thus R-27 and S-27 adopted very different conformations. **b.** $\beta$-NADPH, $\alpha$-NADPH and t-NADPH.

**Table 1. IC$_{50}$ data for R-27, S-27 and TMP.** Adapted from Table 2 in Ref. 15.

| Inhibitor | Sa IC$_{50}$ (nM) | Sa(F98Y) IC$_{50}$ (nM) |
|---|---|---|
| R-27 | 15 ± 0.7 | 510 ± 50 |
| S-27 | 18 ± 2 | 111 ± 6 |
| TMP | 23 ± 3 | 1700 ± 19 |

the ribose appended to nicotinamide. Three forms of NADPH isomers are shown in Fig 2b. Although these observations suggested a close correlation between resistance and cofactor influence, the detailed mechanism behind it remained unknown.

In this paper, we describe how we first refined the X-ray diffraction data of R-27:t-NADPH: SaDHFR (WT) and updated its structure. We reassigned $\alpha$-NAPDH in Ref. 15 as the closed-ring, tricyclic tautomer of NADPH (t-NADPH) as determined by electron density map, and deposited the new structure into the PDB (PDB ID: 7T7S). Using this new structure, we then applied a series of computational tools and methods to elucidate the mechanism of resistance in F98Y SaDHFR mutants.

The major computational tool we used is our free and open-source CPD software suite OSPREY [17]. Given a structural model as input, OSPREY exploits physical chemistry concepts and provable algorithms to calculate a $K^*$ score, which approximates the binding affinity of the given complex [17, 28]. OSPREY also outputs an ensemble of predicted lowest energy conformations of the protein-ligand complex [16, 29, 30], which provides a detailed model to understand the structural basis of binding affinity changes. In this technique, the thermodynamic ensemble is efficiently predicted by OSPREY, and used to compute $K^*$ score which is an approximation of association constant $K_a$. The output molecular ensemble provides a structural model and explanation for the observed experimental data and computational predictions. Herein, computational techniques allowed us to enumerate and fully explore all possible complexes between different PLA enantiomers, NADPH isomers and SaDHFRs (WT vs. F98Y mutant). Excitingly, our computational results showed excellent concordance with $IC_{50}$ data. The $K^*$ scores and ensembles of SaDHFR:cofactor:inhibitor structures calculated by OSPREY suggest an explanation for the stereospecific and cofactor-dependent inhibition of SaDHFR. R-27 and S-27 are an enantiomeric pair of molecules, but when they bind to DHFR they have different preferences for the selection of NADPH isomers. This suggests that there is some form of cooperativity of binding between R-27, S-27 and NADPH, which may be similar to what is described for TMP and NADPH [31, 32]. Moreover, since R-27 and S-27 have differential potency to F98Y SaDHFR (Table 1), we name the phenomenon *chiral evasion*, which occurs when an enzyme exploits the configuration and chirality difference of its cofactor to evade an inhibitor. In most commonly-seen drug resistant systems, resistance-conferring mutations in enzymes ablate binding of inhibitors and this mechanism is independent of chirality (e.g., TMP resistence in F98Y DHFR doesn't involve chirality, since TMP is not a chiral molecule). However, in the chiral evasion case against PLA enantiomers we present in this paper, chirality played an important role in the mechanism of action. There is manifest stereospecific cooperativity between NADPH cofactors and PLA enantiomers, which is supported by crystallographic structure data and the computational results. The OSPREY-produced ensembles of conformations predicted the detailed molecular contacts around the enzyme active sites, providing a structural basis for the mechanism of F98Y-mediated resistance and SaDHFR's chiral evasion. Moreover, our methodology and structural models of resistance mechanism have already been demonstrated to be helpful in a prospective empirical study on developing novel and resilient inhibitors towards drug candidates [33].

The goal in the present paper is to elucidate the structural and biochemical basis of chiral evasion, and inform the design of resilient inhibitors to overcome it. Our results and methodology highlight how computational protein engineering can be used to probe the mechanism of action, illuminate the structural and thermodynamic basis of stereoselective antimicrobial resistance, and aid the design of chiral inhibitors with improved durability. To that end, the following contributions are made in this work:

1. We reanalyzed the crystallography data of R-27:NADPH:SaDHFR complex, and found that an alternative isomer of the NADPH cofactor namely the tricyclic NADPH (t-NADPH) is a better fit to the original data. Therefore, we present a new structure deposition for this new model (PDB ID: 7T7S). It provides our analysis (below) with an accurate structural foundation.

2. We present data from an authentic sample showing that t-NADPH is a species capable of modulating the activity of DHFR.

3. We perform a set of computational analyses to evaluate the interactions of the PLA enantiomers with the protein target.

4. K* scores calculated by OSPREY recapitulate the experimental IC$_{50}$ data and rationalize PLA enantiomers' NADPH configuration preference.

5. We propose a mechanism of F98Y-mediated resistance and SaDHFR's chiral evasion, supported by predicted conformational ensembles.

To clarify the language and terminology used in our manuscript, a table of acronyms and terms of art is presented in Table A in S1 Text.

## Materials and methods

### OSPREY and K* algorithm

OSPREY is an open-source structure based CPD software suite developed by us [17]. For any user-specified mutant sequence of given protein-ligand system, OSPREY is able to calculate a K* score for it which approximates its binding constant K$_a$. K* score is defined as the quotient of bound and unbound partition function (PF) of the given protein-ligand binding system. The proof of theoretical equality between K* and K$_a$ under exact accurate condition can be found in Appendix A in Ref. 29. Bound state refers to the protein:ligand complex, and unbound state is the state when protein and ligand are free and not binding to each other. Let us denote an arbitrary state as $X$, $X \in \{P, L, PL\}$ where $P$, $L$ and $PL$ represent unbound state of protein, unbound state of ligand and bound state of protein-ligand complex, respectively. PF of a given state is a summation of Boltzmann weighted energy value for all conformations in the ensemble of this state. For a given mutant sequence $s$, its PF of state $X$ (which can be denoted as $q_x(s)$) is defined as:

$$\boldsymbol{q}_X(s) = \sum_{c \in \boldsymbol{Q}_X(s)} \exp \frac{-E(c)}{RT} \tag{1}$$

Here $\mathbf{Q}_x(s)$ is the ensemble of state $X$, which consists of every different possible conformation given the sequence $s$. $c$ denotes a single conformation $\mathbf{Q}_x(s)$. $E(c)$ is the conformational energy of $c$. $R$ is ideal gas constant and $T$ denotes absolute temperature. K* score of $s$ then can be written as:

$$K^*(s) = \frac{q_{PL}(s)}{q_P(s)q_L(s)} \tag{2}$$

OSPREY uses K* algorithm to calculate K* scores [29, 30]. The input of K* algorithm includes a starting structure, energy functions, amino acid side chain library and user specified parameters (including flexibility setup, desired mutant sequence, etc.). Under a provable paradigm, it performs A* search [34] over the ensemble and generates an ordered, gap-free list of all low energy conformations.

## Structure preparation and OSPREY setup

OSPREY analysis starts from the input structure, and here are their sources.

In this study, the ternary complex of DHFR:NADPH:PLA is regarded as bound state. For bound state input structures (structures for calculating ($q_{PL}$)):

- R-27:t-NADPH:SaDHFR: 7T7S

- S-27:t-NADPH:SaDHFR: 7T7S, substituted the original ligand R-27 with S-27

- R-27:$\beta$-NADPH:SaDHFR: 5IST, subsituted the original ligand (UCP1106) with R-27

- S-27:$\beta$-NADPH:SaDHFR: 4TU5

Note that 5IST [35] was chosen to model the R-27:$\beta$-NADPH:SaDHFR structure due to the reason that the ligand in 5IST named UCP1106 (also a kind of PLA) adopted a very similar conformation to R-27. Also, for structures containing t-NADPH, 2 key water molecules that bridge contacts near by were included together as well.

For the unbound state structure, since we want to estimate the K$_a$ of inhibitors, we regard the inhibitor by itself as unbound ligand (for $q_L$) and model the binary complex of DHFR: NADPH as unbound receptor protein (for $q_P$). Input structures for $q_P$ came from:

- t-NADPH:SaDHFR: 7T7S, remove ligand R-27

- $\beta$-NADPH:SaDHFR: 4TU5, remove ligand S-27

Input structures for $q_L$ are 3D structures of R-27 and S-27 taken from 7T7S and 4TU5, respectively.

All input structures were separately processed through SANDER minimizer from AMBER tool-box [36, 37] to release some intrinsic minor clashes in crystal structures. OSPREY is then employed to calculated PF and K$^*$ scores. In all of these OSPREY instances, they all have 11 same flexible residues that are Leu5, Val6, Leu20, Asp27, Leu28, Val31, Met42, Thr46, Ile50, Leu54 and Phe92. They have one mutable residue which is Phe98, allowed to mutate to tyrosine, so that we have results for both the WT and F98Y mutant. DHFR, NADPH and inhibitor were modeled with flexibility of translation and rotational movement against each other [38, 39]. The side chain of flexible and mutable residues were modeled with continuous flexibility (side chain is continuous rotatable within a range of ±9˚ from rotamer library's discrete value [19]) as well. For computational efficiency partition functions for each ternary state were calculated with the $3 \times 10^4$ provably lowest-energy conformations, enumerated gap-free.

## Results

### Reassignment of the crystal structure of t-NADPH in complex with DHFR and R-27

NADPH has been observed in different binding states while in complex with SaDHFR and antifolates. In our early study of PLAs in Ref. 12, an alternative configuration of NADPH that is different from the standard form ($\beta$-NADPH) was initially assigned to several crystal structures of WT and F98Y SaDHFR (PDB ID: 3fqf, 3fqo, 3fqv and 3fqz). It was hypothesized that this alternative NADPH configuration could be involved in the drug resistance mechanism of SaDHFR [12]. At that time the alternative form of NADPH was thought to be merely a conformational difference, albeit a unique one. Because of its importance, we then decided to investigate and model this non-standard NADPH retrospectively using our CPD software OSPREY. The OSPREY-based analysis [18] revealed that the alternative conformation of NADPH reported in Ref. 12 actually possesses a different configuration relative to $\beta$-NADPH at the anomeric

center. To investigate more closely we examined the density map of 3fqf, 3fqo, 3fqv, 3fqz, and measured the geometry (i.e., bond angles) of NADPH around the chiral center (C1′ of ribose on nicotinamide side) [18]. The electron density around the ribose sugar of NADPH is extremely well-resolved for both 3fqf and 3fqz, and the best-fit β-NADPH conformation deviated by up to 75˚ from ideal tetrahedral geometry at the anomeric center. This analysis showed conclusively that the epimer in these structures is *not* β-NADPH. To determine the prevalence of this phenomenon we searched over 1700 NADPH-containing PDB structures for similar geometries at the anomeric center. At that time only 23 structures with similar NADPH geometry were found and among them only 3fqf and 3fqz had high enough resolution to conclude anything regarding their anomeric configuration. We also compared to the literature [40–43] reports of an NMR study of an alternative NADPH complexed with *Lactobacillus casei* DHFR (which has the same fold as SaDHFR). Unfortunately, we still found that this alternative NADPH conformation reported in *L. casei* DHFR study [40–43] is distinct from the alternative NADPH we found in our SaDHFR study [12]. In summary, we found that the alternative NADPH in 3fqf, 3fqo, 3fqv, 3fqz in Ref. 12 is in a diastereomeric state, and this analog of NADPH had never been reported to bind with non-SaDHFR or non-PLA inhibitors [18].

In a later study [15] we determined crystal structures of SaDHFR (WT) and NADPH bound to R-27 and S-27, respectively (PDB ID: 4XEC and 4TU5). In particular, R-27 was found to bind to this same alternative NADPH configuration. It was indeed confirmed to be in a different isomeric state from β-NADPH, which validated our analysis in Ref. 18. At that time, the NADPH in R-27 complex was thought to be in its α configuration (α-NADPH, as shown in Fig 2b). However, after new analysis and refinement of the crystal structures we now reveal that the NADPH density is actually in a α-02′-6B-cyclotetrahydronicotinamide adenine dinucleotide configuration (t-NADPH), as shown in Figs 2b and 3.

Further refinement of the SaDHFR:R-27 structure was needed to assess the fit of t-NADPH into the NADPH binding site. To achieve this, both energy minimized α-NADPH and t-NADPH were re-fit into the SaDHFR:R-27 structure using the original electron density map of 4XEC. Upon refinement of these two structures in PHENIX, we observed that the t-NADPH cofactor better satisfied both the local electron density map and difference density that arose in the α-NADPH structure. Additionally, we observe a decrease in the average cofactor B-factors (from 39.98 with α-NADPH to 33.55 with t-NADPH) and the B-factors of the oxygen atom involved in cyclization (from 39.22 with α-NADPH to 34.15 with t-NADPH). Furthermore, there is an overall decrease in $R_{free}$ for the structure with the remediation of the NADPH cofactor, with a decrease from 0.2543 with α-NADPH to 0.2529 with t-NADPH (See Table B in S1 Text), a meaningful decrease given the minor structural differences between cofactor structures. This reduction in $R_{free}$ indicates that t-NADPH better satisfies the electron density map and supports the presence of t-NADPH over the alpha anomer in the active site. Thus, we

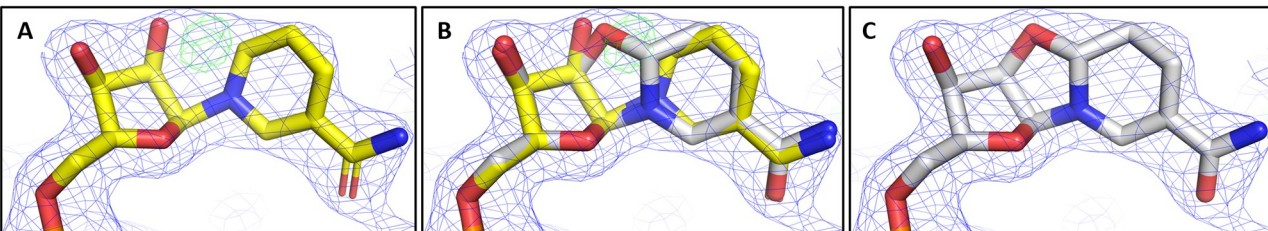

**Fig 3. NADPH anomers in the electron densities. a.** Refinement of α-NADPH (yellow) in the SaDHFR:R-27 crystal structure. Green mesh (difference density) shows unsatisfied electron density. **b.** Overlay of α-NADPH and t-NADPH (light gray) in the SaDHFR:R-27 crystal structure. **c.** Refinement of t-NADPH in the SaDHFR:R-27 crystal structure.

created two new, improved structure models with significant implications in terms of biological physical chemistry (discussed herein). Both new structures were deposited in the PDB. The PDB ID for R-27:$\alpha$-NADPH:SaDHFR is 7T7Q and the PDB ID for R-27:t-NADPH:SaDHFR is 7T7S. Crystallographic data collection and refinement statistics are provided in Table B in S1 Text.

$\beta$-NADPH and $\alpha$-NADPH are both physiologically relevant species [24]. They naturally interconvert although the $\beta$ form is more stable and generally the isomer utilized by DHFR. We have observed via HPLC analysis that $\alpha$-NADPH accounts for 7% of the concentration of NADPH at neutral pH (Table C in S1 Text). However at low pH $\alpha$-NADPH can become trapped as t-NADPH by undergoing a cyclization through the addition of the 20 hydroxyl on the ribose 13 and the tetrahydronicotinamide ring [44, 45]. We also note that the acid treatment of $\beta$-NADPH also leads to t-NADPH via initial anomerization to the $\alpha$ form [45]. To understand the prevalence of $\alpha$-NADPH and subsequent formation of t-NADPH, we used NMR to track the species in biochemical, microbiological and crystallographic conditions (Table C and Figs A-D in S1 Text). Similar to published work [44, 45], under acidic conditions (pH 4.5, RT) we see enrichment of $\alpha$-NADPH to 14% of the total solution and after 7h were able to isolate 30% t-NADPH. Having isolated t-NADPH, we were able to perform biochemical experiments with SaDHFR [15]. We found that alone, t-NADPH is not an active cofactor for SaDHFR. Additionally, we were able to identify that t-NADPH alone has inhibitory effects on SaDHFR with an IC$_{50}$ of 111 $\pm$ 6$\mu$M. Additionally, we have shown that co-incubation of t-NADPH with R-27 and SaDHFR and enzymatic activation with a mix of $\beta$-NADPH and DHF results in a 2-fold reduction in IC$_{50}$ compared to no t-NADPH pre-incubation (Fig E in S1 Text).

One explanation for the presence of t-NADPH is the low pH (pH 6) used during crystallization conditions. Alternatively, the enzyme binding pocket could potentially alter the pKa of NADPH, thus allowing for the cyclization reaction to occur. The biological relevance of t-NADPH, or its very presence in cells is not known so far. However, to date, t-NADPH has only been found with SaDHFR bound to certain antifolates, and more commonly in SaDHFR mutants containing the resistance mutation F98Y. There are other structures of NADPH complexed with DHFR of different species in the PDB that were crystallized at a similarly low pH (PH $\leq$ 6), e.g. 2fzi [46], 3frd [47], 3fy8 [48] etc., but no instances of t-NADPH had ever been found. This suggests that the binding of t-NADPH may play a role in the resistance to antifolates conferred by the F98Y substitution [15]. In the following sections of the paper we use our comprehensive suite of biochemical, structural and computational data to examine the role of t-NADPH.

## Comparison between K$^*$ scores and IC$_{50}$ values

In our previous study [15] we found that SaDHFR, in particular its F98Y mutant, showed remarkable stereospecificity of inhibition (Table 1). IC$_{50}$ values of a series of PLA enantiomer pairs were determined, and crystal structures of the most potent pair S-27 and R-27 complexed with SaDHFR and cofactor NADPH were determined. The crystal structure revealed an interesting phenomenon that S-27 and R-27 bound preferentially with different isomeric states of NADPH. S-27 binds with the most commonly seen form $\beta$-NADPH while R-27 binds with a very rare form t-NADPH. Although the above observations were made, the mechanism of F98Y mediated resistance, the reason of its stereospecificity to enantiomer pairs and the reason for the enantiomers' selectivity to NADPH configurations remained unclear.

To investigate and understand the above questions, we performed computational analysis using our CPD software suite OSPREY. In order to fully cover all the possible situations, we ran

**Table 2. Values of K* scores and partition functions (PF) for different systems calculated by OSPREY.**

| System | Bound state PF ($q_{PL}$) | Unbound state receptor PF ($q_P$) | Unbound state ligand PF ($q_L$) | K* score |
|---|---|---|---|---|
| R27:t-NADPH:DHFR(WT) | 191.15 | 146.05 | 0.30 | 44.79 |
| S27:t-NADPH:DHFR(WT) | 182.38 | 146.05 | 0.30 | 36.02 |
| R27:$\beta$-NADPH:DHFR(WT) | 180.78 | 145.96 | 0.30 | 34.53 |
| S27:$\beta$-NADPH:DHFR(WT) | 190.59 | 145.96 | 0.30 | 44.32 |
| R27:t-NADPH:DHFR(F98Y) | 183.55 | 144.05 | 0.30 | 39.19 |
| S27:t-NADPH:DHFR(F98Y) | 178.82 | 144.05 | 0.30 | 34.27 |
| R27:$\beta$-NADPH:DHFR(F98Y) | 174.80 | 142.29 | 0.30 | 32.22 |
| S27:$\beta$-NADPH:DHFR(F98Y) | 186.30 | 142.29 | 0.30 | 43.71 |

All PF and K* scores in this table are $\log_{10}$ values.

OSPREY on 8 different instances, generated by combinatorially selecting one inhibitor from S-27 or R-27, one NADPH configuration from $\beta$-NADPH or t-NADPH, one enzyme from WT or F98Y DHFR. The generated 8 instances are listed in Table 2. The source of input starting structures and setup parameters for OSPREY are described in the Materials and Methods. The resulting K* scores calculated by OSPREY are shown in Table 2. In these cases, the bound state refers to the ternary complex of DHFR:NADPH:inhibitor. The binary complex of DHFR:NADPH as a whole is regarded as unbound state protein receptor, and the inhibitor by itself is unbound state ligand (as explained in the Materials and Methods).

K* score is an approximation of the binding constant $K_a$. In order to calculate the binding constant for our system, we refer to the usage of partition function (PF), which is a widely used concept in physical chemistry researches. It has been proved in Ref. 29 that K* will be equal to $K_a$ under the condition of using exact PF. K* score is the quotient of the bound and unbound state PF, where PF is a summation of Boltzmann weighted energy value of all conformations in the design ensemble. Let $q$ represent PF, *PL* represent bound state, *P* and *L* represent unbound states protein and ligand, then the definition of K* can be written as $K* = \frac{q_{PL}}{q_P q_L}$ (a more detailed description is presented in the Materials and Methods, and further explanation can be found in Ref. 17, 29, 30). $K_a$ is the reciprocal of the dissociation constant, and we show that it is possible to comparable it with $IC_{50}$ after a simple conversion in this case, although they are not directly equivalent. The Cheng-Prusoff equation describes the relationship between $K_i$ and $IC_{50}$ [49]. The equation for competitive inhibition case (equation 3 in Ref. 49) is:

$$IC_{50} = K_i \left( 1 + \frac{S}{K_m} \right). \tag{3}$$

Here $K_m$ is the Michaelis constant of the substrate and S denotes substrate concentration. For a given type of enzyme and substrate, the value of S and $K_m$ remain constant, thus $K_a$ is proportional to $1/IC_{50}$ ($K_a \propto \frac{1}{IC_{50}}$). According to the previous study, $K_m$ values for WT and F98Y DHFR is measured as 14.5 $\mu$m and 7.3 $\mu$m respectively, and S is approximately 100 $\mu$m [50]. With these data we can convert $IC_{50}$ of our systems into $K_a$, and then compare to K* scores. The result is shown in Fig 4.

From Table 2, we can see that for WT SaDHFR, K* score for R-27:t-NADPH:DHFR(WT) is almost equal to but very slightly higher than for S-27:$\beta$-NADPH:DHFR(WT). They are much higher than K* scores for S-27:t-NADPH:DHFR(WT) and R-27:$\beta$-NADPH:DHFR (WT). Such result agrees with $IC_{50}$ data very well. It also indicates that R-27 complexed with t-NADPH is more energetically favorable than when complexed with $\beta$-NADPH, and on the

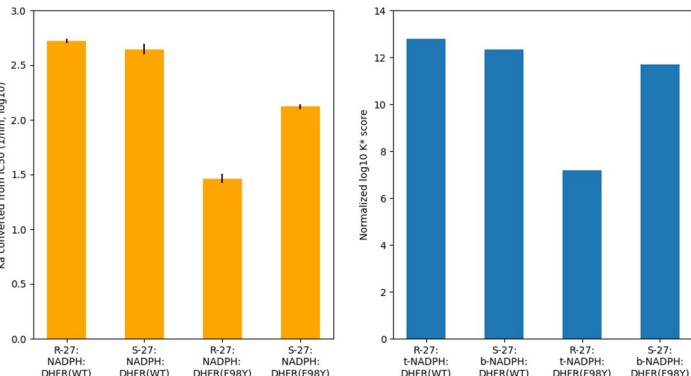

**Fig 4. Comparison of $K_a$ (derived from $IC_{50}$ data) and normalized $\log_{10} K^*$ score.** Bar graphs of $K_a$ and $K^*$ score values (both in $\log_{10}$) for different systems. $K^*$ scores are normalized as described in S1 Text. Normalized $\log_{10} K^*$ scores are in good concordance with $K_a$, which is derived from $IC_{50}$ using Cheng-Prusoff equation. Black bars represent uncertainty from experimental measurments of $IC_{50}$.

contrary S-27 is more energetically favorable when complexed with $\beta$-NADPH than when complexed with t-NADPH. This can explain the difference in NADPH isomeric state observed in crystal structures.

For SaDHFR F98Y, although no crystal structure bound with R-27 or S-27 has been determined, we made homology models (as described in the Materials and Methods) to investigate their properties. Based on our models, OSPREY predicts that R-27 and S-27 demonstrate the different NADPH configuration preference, as they did when bound with WT SaDHFR. Similarly to those with WT DHFR, $K^*$ scores for R-27:t-NADPH:DHFR(F98Y) and S-27:$\beta$-NADPH: DHFR(F98Y) are higher than those for S-27:t-NADPH:DHFR(F98Y) and R-27:$\beta$-NADPH: DHFR(F98Y). Moreover, $K^*$ scores for S-27 bound with F98Y DHFR only decreased to a moderate extent, compared to when binding with WT DHFR (the $\log_{10} K^*$ score for S-27:$\beta$-NADPH:DHFR(WT) is 44.32 and for S-27:$\beta$-NADPH:DHFR(F98Y) is 43.71, decrease in $\log_{10}$ $K^*$ score between WT and F98Y is 0.61). But for R-27, compared with bound with WT DHFR, $K^*$ score decreased when binding with F98Y (the $\log_{10} K^*$ score for R-27:t-NADPH:DHFR (WT) is 44.79 and for R-27:t-NADPH:DHFR(F98Y) is 39.19, decrease in $\log_{10} K^*$ score between WT and F98Y is 5.60). To make a comparison between $K^*$ scores with $IC_{50}$ data and determine their concordance, we normalized the $K^*$ scores as described in S1 Text. These normalized $K^*$ scores showed remarkably good concordance with $IC_{50}$ data (see Fig 4). Based on these results, our computational analysis successfully recapitulated the experimental thermodynamic and structural data.

## Structural analysis

OSPREY can predict and calculate not only the $K^*$ score for each protein/ligand complex, but also an ensemble that consists of a series of conformations ranked in the order of energy. These ensembles and conformations exhibit very detailed contact information around the active site, and examining these structures helps us understand the structural basis of F98Y resistance as well as the stereospecific inhibition of DHFR.

Lowest energy conformations from ensembles of 4 wild type SaDHFR-related complexes (R-27:t-NADPH:DHFR(WT), R-27:$\beta$-NADPH:DHFR(WT), S-27:$\beta$-NADPH:DHFR(WT) and S-27:t-NADPH:DHFR(WT)) are shown in Fig 5, respectively. The interaction between antifolates (R-27 or S-27) and DHFR and NADPH is visualized using Probe dots [51]. Red and yellow dots represent unfavorable overlap, and green and blue dots represent H-bonds and van

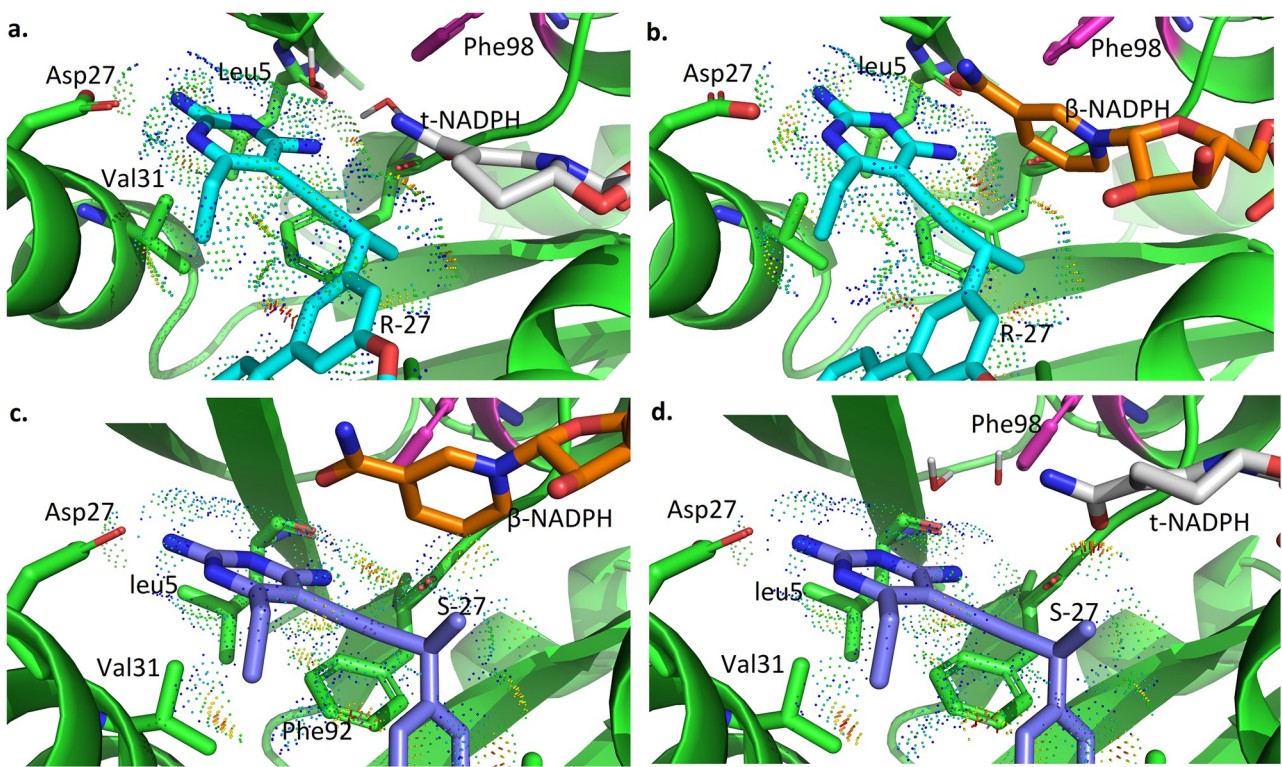

**Fig 5. OSPREY-predicted low energy conformations for ternary complexes. a.** R27:t-NADPH:DHFR(WT) **b.** R27:$\beta$-NADPH:DHFR(WT) **c.** S27:$\beta$-NADPH:DHFR(WT) **d.** S27:t-NADPH:DHFR(WT).

der Waals (vdW) contacts. In all of our analyses, two water molecules in the binding pocket were modeled along with t-NADPH in all complexes containing t-NADPH, since these water moelcules are crucial for bridging contact between t-NADPH, antifolates and DHFR.

As shown in Fig 5, in R-27:t-NADPH:DHFR(WT) system, there are many favorable interactions between R-27 and t-NADPH (panel a). Replacing t-NADPH with $\beta$-NADPH (panel b) produces some steric clashes between $\beta$-NADPH and R-27, since $\beta$-NADPH extends deeper in the binding pocket relative to t-NADPH. It suggests that R-27 binding with $\beta$-NADPH is less thermodynamically stable than binding with t-NADPH, thus we saw a higher K* score for R-27 binding to t-NADPH and they showed a 100% occupancy in the crystal structure. For S-27, there are more favorable interactions between S-27 and $\beta$-NADPH (panel c). However, when t-NAPDH is modeled in the same location (panel d), it diminishes many of these favorable contacts. Therefore, S-27 is more stable with $\beta$-NADPH, and this could explain why S-27 prefers a different NADPH configuration relative to R-27.

To better understand these stereochemical preferences, we performed a structural comparison between the WT and F98Y DHFR binding to R-27 (together with t-NADPH) and S-27 (together with $\beta$-NADPH) (see Figs 6 and 7, respectively). In R-27 related complexes (Fig 6), compared with the WT DHFR (panel a), the F98Y mutation introduces steric clashes between Tyr98 and Gly93 (panel b). Since tyrosine is bulkier than phenylalanine in size, its side chain is hard to fit in the very confined space around the active site. Clashes induced by F98Y mutation mainly happened between the side chain of Tyr98 and the backbone of Gly93, making it energetically unfavorable. Here the closet distance between Tyr98 and Gly93 is 2.3 Å, which is much closer than what is expected for ideal van der Waals contacts. Panel c and d in Fig 6 showed the lowest energy conformation in binary complex ensembles of t-NADPH:DHFR

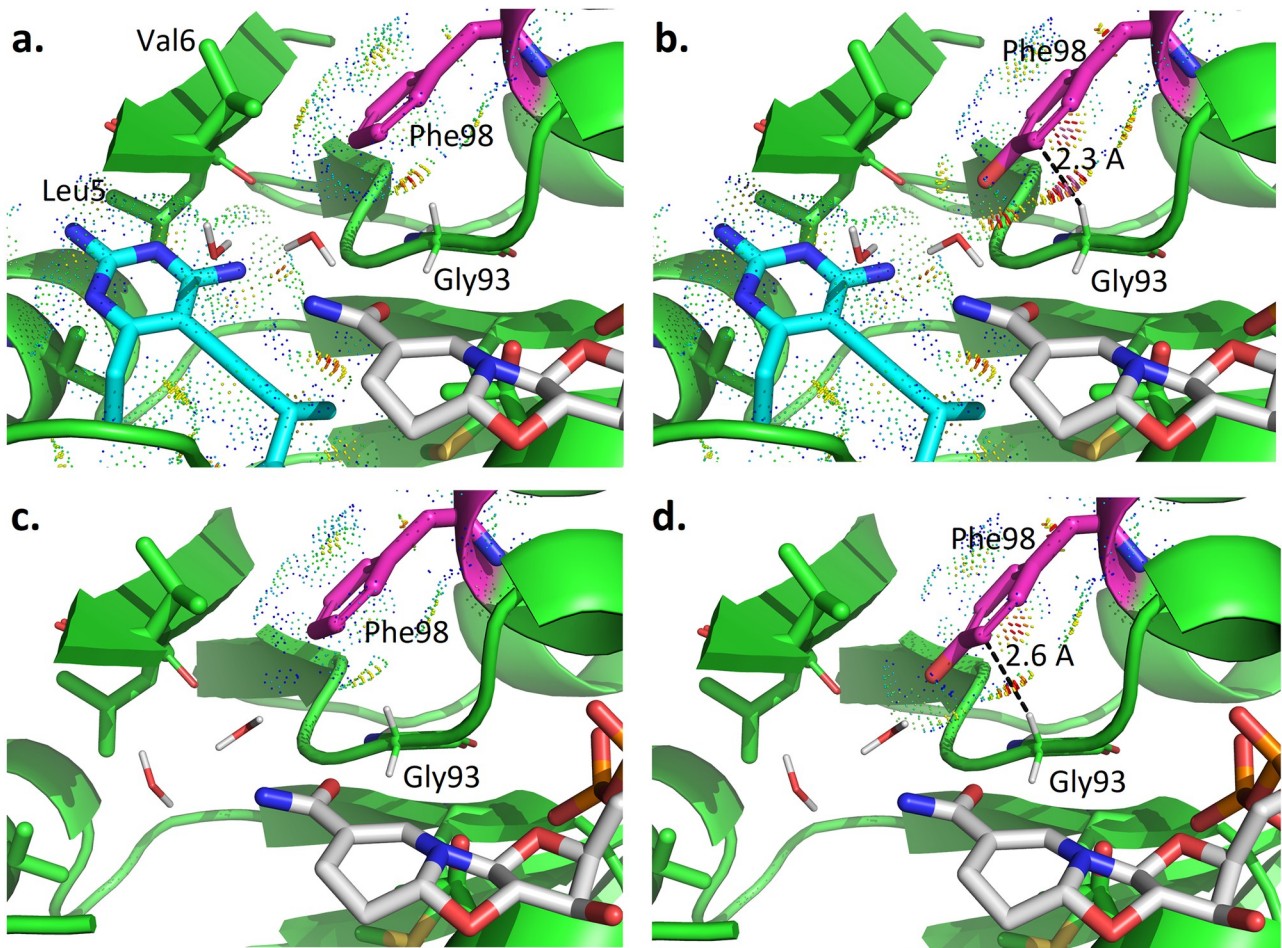

**Fig 6. OSPREY-predicted low energy conformations for t-NADPH related complexes. a.** Ternary complex of R27:t-NADPH:DHFR(WT) **b.** Ternary complex of R27:t-NADPH:DHFR(F98Y) **c.** Binary complex of t-NADPH:DHFR(WT) **d.** Binary complex of t-NADPH:DHFR(F98Y).

(WT) and t-NADPH:DHFR(F98Y). According to these binary complex conformations, when no inhibitor binds, space in the pocket is left unoccupied. The two water molecules mediating molecular contact for t-NADPH then tend to move outwards to fill up the empty space. Consequently, clashes between Tyr98 and Gly93 that resulted from the crowded space in the ternary complex can be relieved to a large extent in the binary complex. The distance between Tyr98 and Gly93 increases from 2.3 Å (Fig 6b) to 2.6 Å (Fig 6d). Additionally, inspection of an ensemble of the 30 lowest-energy conformations for t-NADPH -related binary complexes reveals a minor population characterized by small (intra-rotamer) changes to residue 98 and a corresponding change in rotamer of residue V6 (Fig F in S1 Text). The existence of this minor population suggests that the active sites of t-NADPH binary complexes may be less tightly-packed than all other modeled states (i.e. ternary states and $\beta$-NADPH binary complexes), for which no minor populations were predicted at these residues. The energy and binding affinity change resulted from such difference can be reflected by PF and K* scores. As we can see in Table 2, the bound state PF ($\log_{10}$) value of R-27:t-NADPH:DHFR(F98Y) is much lower than R-27:t-NADPH:DHFR(WT) (183.55 vs. 191.15 in $\log_{10}$ value), but their unbound state PF value of is quite close (144.05 and 146.05 in $\log_{10}$ value). Therefore, their K* scores (as the quotient of bound and unbound state PF) differ (39.19 vs. 44.79 in $\log_{10}$ K* value).

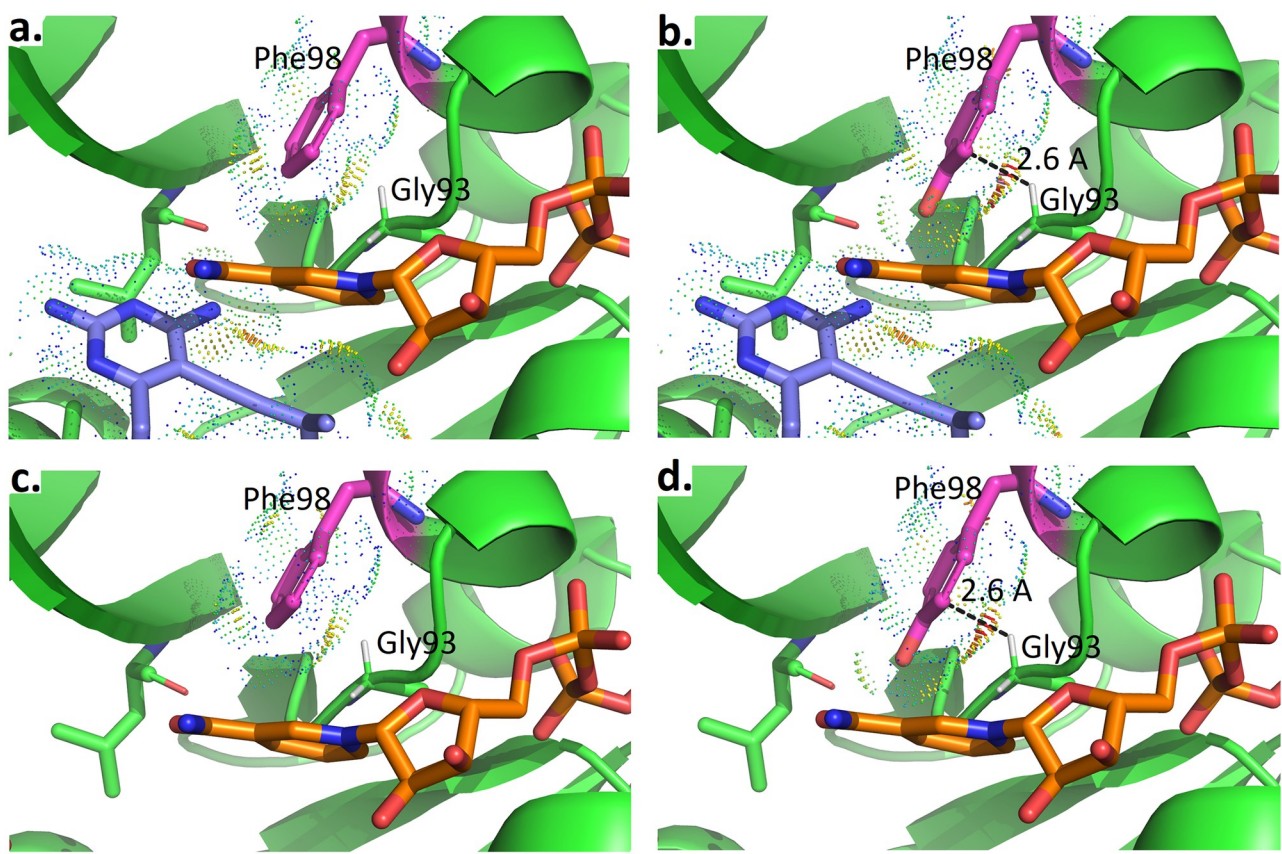

**Fig 7. OSPREY-predicted low energy conformations for β-NADPH complexes. a.** Ternary complex of S27:β-NADPH:DHFR(WT) **b.** Ternary complex of S27:β-NADPH:DHFR(F98Y) **c.** Binary complex of β-NADPH:DHFR(WT) **d.** Binary complex of β-NADPH:DHFR(F98Y).

For S-27 binding complexes (Fig 7), the influence of the F98Y mutation (panel a and b) is very similar to that with R-27. Compared with the WT amino acid at residue 98 (phenylalanine), the mutation to tyrosine leads to clashes with Gly93. However, the binary complex conformations for β-NADPH:DHFR(WT) and β-NADPH:DHFR(F98Y) (panel c and d) are very different with those with t-NADPH. β-NADPH itself forms substantial contact with DHFR and unlike t-NADPH, there isn't any water involved in bridging interactions. Even in the binary complex, β-NADPH is very restricted such that clashes between Phe98 and Gly93 cannot be relieved. Quantitatively, comparing the WT and F98Y DHFR binding to S-27 and β-NADPH, PF of both bound and unbound states decrease by similar amounts. Bound state PF values of S27:β-NADPH:DHFR(WT) and S27:β-NADPH:DHFR(F98Y) are 190.59 and 186.30 respectively, and unbound state PF are 145.96 and 142.29, all in $\log_{10}$. Since K* score is the quotient of bound and unbound state PF, when bound and unbound state PF decrease by a similar amount, the decrease in unbound state PF will compensate for the decrease in bound state PF. As a result, ($\log_{10}$) K* score did not change too much (44.32 for S27:β-NADPH:DHFR(WT) and 43.71 for S27:b-NADPH:DHFR(F98Y)).

## Discussion

Investigations of F98Y-mediated resistance in SaDHFR point to the importance of the NADPH cofactor-binding site and cofactor configuration to inhibitor binding. Crystal structures of various competitive inhibitors of SaDHFR [12, 15, 35] revealed an alternative form of

NADPH that appeared in complex with inhibitors both with and without the F98Y resistance mutation. This alternative NADPH was initially believed to be an alternative conformation [12], was later assigned as $\alpha$-NADPH [15, 18, 35], and in this manuscript has been identified as t-NADPH in the structure of SaDHFR bound to the R-enantiomer of compound 27. The alternate cofactor appeared in a ligand-dependent manner [12], and its incidence was higher overall in structures of SaDHFR(F98Y). Later, the alternate cofactor appeared in a stereospecific manner in structures of enantiomers of compound 27 bound to wild-type SaDHFR [15]. Finally, structures of inhibitors bound to SaDHFR both with and without the F98Y mutation indicated that the alternate cofactor may play a role in stabilizing the hydrogen bond between inhibitors and the carbonyl of residue F92 [35]. These data, along with the observation that F98Y disrupts cooperativity of binding between NADPH and TMP [52], suggest that F98Y may mediate resistance by perturbing the cofactor binding site.

In this manuscript we investigate the mechanism behind the F98Y-induced stereospecificity of inhibition by enantiomers of compound 27 [15]. In doing so, we provide additional evidence for the importance of the cofactor-binding site and cofactor configuration to F98Y-mediated resistance for SaDHFR. We show that the cofactor present in the crystal structure of SaDHFR:R-27 is t-NADPH, we show that t-NADPH can inhibit SaDHFR and interact cooperatively with R-27 *in vitro*, and we provide computational modeling results that suggest that F98Y affects the distribution of cofactor states. Finally, we suggest a structural and thermodynamic model for the basis of F98Y-induced stereospecificity in this system (Fig 8).

The cofactor electron density for the structure of SaDHFR:R-27 is better-satisfied by assigning t-NADPH to the cofactor-binding site. Computing difference density based on the original assignment of $\alpha$-NADPH revealed an unsatisfied region of density at the site of ring-closure, which is satisfied by reassignment of t-NADPH (Fig 3). This reassignment also results in global decreases in $R_{free}$ and decreases in cofactor B-factors, indicating that the reassignment better satisfies the experimental data. We also considered the possibility that the density results from the averaging of a broad ensemble of $\beta$-NADPH conformations, but discarded this hypothesis due to its complexity and the well-resolved density around the cofactor. The discovery of an alternate form of NADPH binding to SaDHFR suggests that multiple forms of NADPH, including $\alpha$-NADPH and t-NADPH, could be considered as candidates for structures in which electron density does not allow fitting of typical $\beta$-NADPH conformations, and additionally supports the relevance of alternate cofactor configurations in this system.

Results from *in vitro* assays of SaDHFR activity in the presence of t-NADPH support the claim that t-NADPH can bind SaDHFR. We were unsurprised to find that t-NADPH is not a catalytically-active cofactor of SaDHFR, but did find that t-NADPH itself has weak inhibitory effects on SaDHFR ($IC_{50}$: 111 ± 6 $\mu$M, see S1 Text). We also showed that incubation with both t-NADPH and R-27 results in an approximately two-fold decrease in $IC_{50}$ compared to incubation with R-27 alone (Fig E in S1 Text). Together these experiments support the claim that t-NADPH can bind to SaDHFR and in particular may bind cooperatively with R-27. Crystallographic and experimental data indicate that bound states of SaDHFR containing alternative configurations of NADPH can be induced in a ligand-dependent manner.

Computational models of F98Y-induced stereospecific inhibition generated using OSPREY recapitulated crystallographic and experimental trends. We made models of ternary bound states for combinations of cofactor ($\beta$-NADPH, t-NADPH), inhibitor (S-27, R-27), and enzyme (WT, F98Y), and additionally modeled binary states for combinations of cofactor ($\beta$-NADPH, t-NADPH) and enzyme (WT, F98Y). We computed approximations to $K_a$ using the $K^*$ algorithm in OSPREY [17, 29, 30] for each of these states. Our modeling recapitulated the cofactor preferences exhibited in the crystal structures of compound 27 enantiomers: The SaDHFR(WT):t-NADPH:R-27 state was predicted to be the most stable ternary complex of

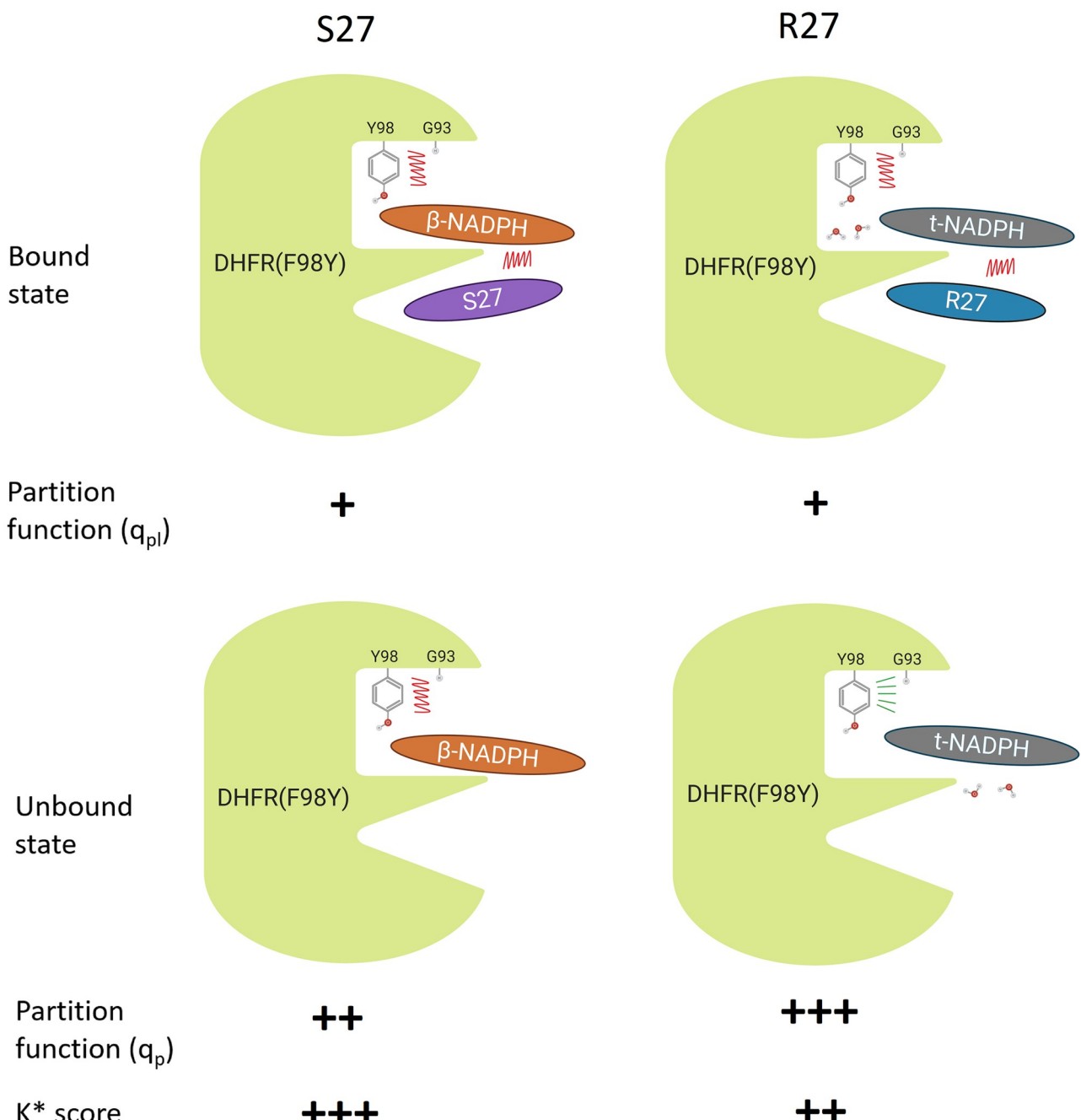

**Fig 8. Proposed mechanism for chiral evasion of SaDHFR and resistance resilience by PLAs.** R-27 and S-27 have almost equal potency when bound to WT DHFR. However, S-27 binds significantly more tightly to F98Y DHFR than R-27 does. The different configuration of NADPH cofactor played an important role, which we term chiral evasion, according to our data. F98Y mutation introduces steric clashes (red jagged lines), and thus makes the whole complex energetically less favorable. By exploiting chiral inhibitor design, an inhibitor (S-27) from a PLA enantiomer pair was obtained that is resilient to chiral evasion. The opposite anomer R-27 is significantly less resilient. When R-27 binds to F98Y DHFR, it binds together with t-NADPH and also two water molecules that bridge the interaction. According to the OSPREY predictions, in the unbound state, the two water molecules move away from the binding pocket and move towards the space left by the absence of the inhibitor. Consequently, the crowding in the binding pocket is alleviated and the clashes can be relieved (green radial lines) in the unbound state. In contrast, when S-27 binds to F98Y DHFR, it only binds together with $\beta$-NADPH (without any water molecules) and the clashes cannot be relieved in the unbound state. Therefore, as a quotient of bound and unbound state partition function, the K* score (which approximates $K_a$) of S-27 is higher than that of R-27. This suggests S-27 should bind tighter to F98Y DHFR than R-27 does; furthermore this prediction agrees with $IC_{50}$ data. Design for resilience to chiral evasion can therefore overcome drug resistance in a protein target.

R-27, and the SaDHFR(WT):$\beta$-NADPH:S-27 state was predicted to be the most stable ternary complex of S-27 (Table 2). Furthermore, we modeled the effects of the F98Y mutation and recapitulated $IC_{50}$ measurements [15] with high accuracy (Fig 4). The excellent concordance with experimental data suggests that these computational models may be capturing biochemically-relevant information. Notably, exclusion of thermodynamic states involving t-NADPH from our models (i.e. considering only the states involving $\beta$-NADPH) prevents recapitulation of experimental $IC_{50}$ measurements, indicating the importance of these states to our model.

Structural ensembles generated from our *in silico* models suggest a mechanism for F98Y-induced stereospecificity of inhibition in this system. In general, the F98Y mutation induces crowding of the active site and clashes with the Gly93 loop, decreasing the partition function for most modeled states (Figs 6 and 7, and Table 2). This is consistent with the previous observation that the F98Y mutation can induce a rotation of the Phe92 carbonyl oxygen, which can in some cases result in the loss of a hydrogen bond with inhibitors [35]. This F98Y-induced crowding can be only relieved in the SaDHFR(F98Y):t-NADPH binary state (Fig F in S1 Text). Additionally, the F98Y mutation adds a hydroxyl group that can participate in a favorable water network when t-NADPH is present in the active site, a phenomenon that has been observed previously in crystal structures [12, 35]. As a result, our models predict that the F98Y mutation increases the favorability of the t-NADPH binary state, while decreasing the favorability of ternary states and the $\beta$-NADPH binary state (Table 2). Because ligand binding depends on a ratio of bound and unbound states, this thermodynamic and structural model predicts that F98Y causes a disproportionate decrease in the binding of R-27 while affecting S-27 to a much lesser extent (Fig 8). This thermodynamic mechanism is consistent with experimental data and provides a hypothesis to explain the unusual phenomenon of F98Y-induced stereospecific inhibition.

This proposed mechanism also prompts several interesting questions regarding the *in vivo* relevance of t-NADPH and the role of kinetic effects in this system. Although the crystallographic and *in vitro* evidence supporting the relevance of t-NADPH to this system is clear, the role of t-NADPH in cells is unknown. We propose several possibilities: Our examination of populations of $\alpha$-NADPH and t-NADPH by HPLC and NMR under various conditions indicate that trace amounts of t-NADPH can be observed under physiologically-relevant conditions (Table C in S1 Text). Alternatively, given the structural similarity between $\alpha$-NADPH and t-NADPH, it is plausible that $\alpha$-NADPH could fill the role of t-NADPH in this mechanism. Due to the rapid interconversion between $\alpha$-NADPH and $\beta$-NADPH we could not perform binding experiments with pure samples of $\alpha$-NADPH. It is also possible that anomerization or cyclization could occur within the enzyme *in situ* in solution, or perhaps only under crystallographic conditions. However, due to the experimental evidence herein supporting t-NADPH binding to SaDHFR (S1 Text and Fig E in S1 Text), we believe this last possibility to be less likely. Further study will be required to conclusively assign the role of t-NADPH *in vivo*. Finally, previous work has suggested that the F98Y mutation may affect binding kinetics [15]. The computational modeling techniques presented here provide thermodynamic information, and alternative approaches will be required to probe the kinetics of this system.

## Conclusion

Previous experiments showed that F98Y SaDHFR, a TMP-resistant mutant of SaDHFR, is stereospecific to a pair of enantiomeric PLAs R-27 and S-27. In this study, our CPD software suite OSPREY was employed to explore the mechanism of this stereospecific inhibition, namely, SaDHFR's chiral evasion against PLA enantiomers. Compared to $IC_{50}$ data, K* scores (which

predict K$_a$) produced by OSPREY successfully recapitulated the ranking of PLA enantiomers' affinity and the ranking of the impact the F98Y mutation would have in the interaction. Among all complexes, K* scores for R-27:t-NADPH:SaDHFR and S-27:$\beta$-NADPH:SaDHFR are significantly higher than for all other models (Table 2), which is consistent with NADPH configuration preferences observed in crystal structure (R-27 bound with t-NADPH and S-27 bound with $\beta$-NADPH, as seen in models 7T7S and 4TU5). Ensembles of conformations of F98Y SaDHFR binding to R-27 and S27 were predicted as well. Based on our structural analysis, we found that different binding modes between t-NADPH (which R-27 prefers) and $\beta$-NADPH (which S-27 prefers) are likely to be a key factor of chiral evasion. The major difference between t-NADPH and $\beta$-NADPH is how they interact with Gly93 loop on SaDHFR. Such difference may lead to clashes between Gly93 and Tyr98 in F98Y mutant, and thus may influence the thermodynamic stability of SaDHFR:NADPH binary complex, ultimately modulating the inhibition potency of R-27 and S-27.

In conclusion we showed how the discovery of a configuration change in NADPH can elucidate a potential mechanism of drug resistance in SaDHFR. This study suggests that the cofactor stereogenicity and chiral evasion should be taken into account when designing new drugs for F98Y SaDHFR. The importance of chirality cannot be revealed by merely studying traditional antifolates such as TMP. The use of computational drug design and protein design algorithms provided new insight, informed hypotheses in biology, and made contributions to biochemistry. The data and models we present here have already been useful in medicinal chemistry campaigns for F98Y resilient inhibitors [33], which suggests that they will have significant value for the scientific community.

## Supporting information

**S1 Text. Supplementary information.** Contains a list of acronyms used in this manuscript, crystallographic data collection and refinement statistics, descriptions of and data on biochemical experiments for NADPH, supplementary structure figures, and a discussion of the normalization and conversion between K* scores and IC$_{50}$. Table A. List of acronyms used in this manuscript. Table B. Crystallographic structure data collection and refinement statistics for 7T7Q and 7T7S. Table C. Equilibrium ratios of $\alpha/\beta$-NADPH and isolated yields of t-NADPH under various conditions. Fig A. $^1$H NMR shows acid-mediated transformation of $\beta$-NADPH to t-NADPH. Fig B. Diastereomers observed via $^1$H NMR from the cyclization reaction of NADPH. Fig C. $^1$H NMR shows thermal isomerization at 100˚C. Fig D. HPLC traces of $\beta$-NADPH, $\alpha$-NADPH, and t-NADPH. Fig E. Pre-incubation with t-NADPH increases the inhibition potency of R-27. Fig F. Minor populations in low-energy ensemble of t-NADPH binary complexes.
(PDF)

**S1 Data. $^1$H NMR Spectra of t-NADPH.**
(PDF)

**S2 Data. $^{13}$C NMR Spectrum.**
(PDF)

**S3 Data. COSY NMR Spectrum.**
(PDF)

**S4 Data. NOESY NMR Spectrum.**
(PDF)

**S5 Data. HSQC NMR Spectra.**
(PDF)

**S6 Data. $^{31}$P NMR Spectrum.**
(PDF)

**S7 Data. $^{1}$H NMR Spectrum of deuterated t-NADPH.**
(PDF)

**S8 Data. HSQC NMR Spectra of deuterated t-NADPH.**
(PDF)

## Acknowledgments

We thank all the Donald lab and the Wright lab members, Prof. Terrence Oas, Prof. Pei Zhou and Prof. David Richardson for their helpful discussion. Fig 8 is created with BioRender.com.

## Author Contributions

**Conceptualization:** Pablo Gainza, Vance G. Fowler, Dennis L. Wright, Bruce R. Donald.

**Funding acquisition:** Dennis L. Wright, Bruce R. Donald.

**Investigation:** Siyu Wang, Stephanie M. Reeve, Graham T. Holt, Adegoke A. Ojewole, Marcel S. Frenkel, Santosh Keshipeddy.

**Resources:** Dennis L. Wright, Bruce R. Donald.

**Software:** Siyu Wang, Graham T. Holt, Adegoke A. Ojewole, Bruce R. Donald.

**Supervision:** Dennis L. Wright, Bruce R. Donald.

**Writing – original draft:** Siyu Wang, Bruce R. Donald.

**Writing – review & editing:** Graham T. Holt, Bruce R. Donald.

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
