## [Decision Letter · Decision Letter 0]

11 Sep 2021

Dear Dr. Donald,

Thank you very much for submitting your manuscript "Chiral Evasion and Stereospecific Antifolate Resistance in Staphylococcus Aureus" for consideration at PLOS Computational Biology.

As with all papers reviewed by the journal, your manuscript was reviewed by members of the editorial board and by several independent reviewers. In light of the reviews (below this email), we would like to invite the resubmission of a significantly-revised version that takes into account the reviewers' comments.

We cannot make any decision about publication until we have seen the revised manuscript and your response to the reviewers' comments. Your revised manuscript is also likely to be sent to reviewers for further evaluation.

Sincerely,

Joanna Slusky, Ph.D.

Guest Editor

PLOS Computational Biology

Nir Ben-Tal

Deputy Editor

PLOS Computational Biology

Reviewer's Responses to Questions

**Comments to the Authors:**

Reviewer #1: the review is uploaded as an attachment.

Reviewer #2: Chiral evasion and stereospecific antifolate resistance in Staphylococcus aureus

(PCOMBIOL-D-21-01260)

In this study, authors have used computational protein design (CPD) software suite OSPREY to explore the mechanism of this stereo-specific inhibition, namely, Staphylococcus aureus dihyrofolate reductases (SaDHFR's) chiral evasion against PLAs (propargyl-linked antifolates) enantiomers. Compared to IC50 data, K* scores (which predict Ka) produced by OSPREY successfully recapitulated the ranking of PLA enantiomers' afinity and the ranking of the impact the F98Y mutation would have in the interaction. Among all complexes, K* scores for R-27:t-NADPH:SaDHFR and S-27:β-NADPH:SaDHFR are significantly higher than for all other models, which is consistent with NADPH configuration preferences observed in crystal structure (R-27 bound with t-NADPH and S-27 bound with β-NADPH, as seen in models 6wmy and 4tu5). Ensembles of conformations of F98Y SaDHFR binding to R-27 and S-27 are predicted as well in the present study. Based on structural analysis, authors have found that different binding modes between t-NADPH (which R-27 prefers) and β-NADPH (which S-27 prefers) are likely to be a key factor of chiral evasion. Authors have shown major difference between t-NADPH and β-NADPH is how they interact with Gly93 loop on SaDHFR. Such difference may lead to clashes between Gly93 and Tyr98 in F98Y mutant, and thus may influence the thermodynamic stability of SaDHFR:NADPH binary complex, ultimately modulating the inhibition potency of R-27 and S-27.

In this whole study authors have shown how the discovery of a configuration change in NADPH can elucidate a potential mechanism of drug resistance in SaDHFR. This study suggests that the cofactor stereogenicity and chiral evasion should be taken into account when designing new drugs for F98Y SaDHFR. The use of computational drug design and protein design algorithms gained new insight, informed hypotheses in biology, and made contributions to biochemistry. The data and models authors presented in this manuscript have already been useful in medicinal chemistry campaigns for F98Y resilient inhibitors, which suggest that they will have significant value for the scientific community. I believe this is significant advancement compare to previous work from the same group and I strongly recommend for its publication.

Reviewer #3: In this manuscript, the authors combine structural analysis with simulations to propose an interesting antimicrobial resistance mechanism through “chiral evasion” from the F98Y mutant of SaDHFR. The manuscript is well written and presented.

First, I must say that I am not well versed in X-ray crystallography and so I cannot comment on or critique this aspect of the work. I do wonder though whether the reported change in the Rfree value from 0.2543 to 0.2529 when going from α-NADPH to t-NADPH should be considered a significant improvement though?

I can however say that the simulations and subsequent simulation analysis look well performed. I’m confident this article would be of interest to the readership and I would like to recommend it for publication after addressing the following minor concerns:

1. The OSPREY software is publicly available yes, but the scripts used by the authors along with the preprocessed input structures could be shared to aid reproducibility.

2. It would be instructive to provide error estimates on the bar chats for Figure 3.

3. The label sizes on Figures 4-6 should be increased.

4. Can the authors comment on the variability of the results obtained for different snapshots from their ensemble with the Probe dots analysis. It would be good to confirm the lowest energy conformer can adequately represent the ensemble obtained from OSPREY. I mean in terms of the subsequent insights taken from the Probe dots calculations.

5. Some very minor corrections I spotted along the way:

“It has been proved in Ref. 30 that K* will be equal to Ka under the condition of using exact” (add “be”).

“Under this model, the ability of DHFR to readily form binary complexes…” (replace from with form).

**Have the authors made all data and (if applicable) computational code underlying the findings in their manuscript fully available?**

Reviewer #1: Yes

Reviewer #2: None

Reviewer #3: **No: **The scripts used by the authors along with the preprocessed input structures could be shared to aid reproducibility. (same as my minor comment 1).

PLOS authors have the option to publish the peer review history of their article (what does this mean?). If published, this will include your full peer review and any attached files.

Reviewer #1: No

Reviewer #2: No

Reviewer #3: **Yes: **Rory Maurice Crean
---

## [Decision Letter · Decision Letter 1]

21 Jan 2022

Dear Dr. Donald,

We are pleased to inform you that your manuscript 'Chiral Evasion and Stereospecific Antifolate Resistance in Staphylococcus Aureus' has been provisionally accepted for publication in PLOS Computational Biology.

Best regards,

Joanna Slusky, Ph.D.

Associate Editor

PLOS Computational Biology

Nir Ben-Tal

Deputy Editor

PLOS Computational Biology

Reviewer's Responses to Questions

**Comments to the Authors:**

Reviewer #1: review is attached

Reviewer #3: I thank the authors for taking the time to respond to my questions and making the extra changes. I am happy with their responses.

**Have the authors made all data and (if applicable) computational code underlying the findings in their manuscript fully available?**

Reviewer #1: Yes

Reviewer #3: Yes

PLOS authors have the option to publish the peer review history of their article (what does this mean?). If published, this will include your full peer review and any attached files.

Reviewer #1: No

Reviewer #3: No

---

## [Editor Report · Acceptance letter]

7 Feb 2022

PCOMPBIOL-D-21-01260R1 

Chiral Evasion and Stereospecific Antifolate Resistance in *Staphylococcus Aureus*

Dear Dr Donald,

I am pleased to inform you that your manuscript has been formally accepted for publication in PLOS Computational Biology. Your manuscript is now with our production department and you will be notified of the publication date in due course.

With kind regards,

Anita Estes
